# Unraveling Youth Trauma and Parental Influence After Twin Earthquakes

**DOI:** 10.3390/healthcare13111249

**Published:** 2025-05-26

**Authors:** Georgios Giannakopoulos, Foivos Zaravinos-Tsakos, Ignatia Farmakopoulou, Bjorn J. van Pelt, Athanasios Maras, Gerasimos Kolaitis

**Affiliations:** 1Department of Child Psychiatry, School of Medicine, National and Kapodistrian University of Athens, Aghia Sophia Children’s Hospital, Thivon & Papadiamantopoulou, 115 27 Athens, Greece; foivoszartsa@med.uoa.gr (F.Z.-T.); gkolaitis@med.uoa.gr (G.K.); 2Department of Education and Social Work, University of Patras, Archemedes Str., Building 7, Rion, 265 04 Patras, Greece; ifarmakop@upatras.gr; 3Department of Child and Adolescent Psychiatry/Psychology, Erasmus MC-Sophia, Dr. Molewaterplein 40, 3015 GD Rotterdam, The Netherlands; b.vanpelt@erasmusmc.nl; 4ARQ National Psychotrauma Center, Nienoord 5, 1112 XE Diemen, The Netherlands; a.maras@arq.org

**Keywords:** earthquakes, children and adolescents, posttraumatic stress disorder (PTSD), parental psychopathology, social support

## Abstract

**Background**: Earthquake exposure has been linked with high rates of posttraumatic stress symptoms (PTSS) and comorbid conditions. Familial factors play critical roles in modulating these outcomes. This study examined youth trauma and parental influence following the twin earthquakes in Kefalonia, Greece, in 2014; **Methods**: A cross-sectional study was conducted with 502 adolescents (aged 11–18 years) and 474 parents from three regions categorized by proximity to the earthquake epicenter. Standardized self-report measures were administered. Data were analyzed using descriptive statistics, correlation analyses, and multiple hierarchical regression analyses to identify key predictors of adverse outcomes; **Results**: Among children, 5.2% exhibited probable PTSD, with girls reporting significantly higher symptom levels than boys. Higher earthquake exposure was associated with elevated PTSS and anxiety. In parents, 44.3% met criteria for probable PTSD, and those in the epicenter group reported significantly higher levels of stress, anxiety, and sleep disturbances. Earthquake exposure was identified as the strongest predictor of adverse outcomes, with parental psychopathology and diminished social support further contributing to increased symptom severity in children; **Conclusions**: The study demonstrates that both direct earthquake exposure and familial factors—particularly parental mental health and social support—play critical roles in shaping posttraumatic outcomes in youth, underscoring the need for integrated, family-centered mental health interventions in post-disaster settings.

## 1. Introduction

Earthquakes are catastrophic events that generate significant physical and psychological trauma, due to their sudden onset and extensive destruction. Such events not only cause immediate harm but also instill a persistent fear of future disasters [1]. This is particularly evident in children and adolescents, who often lack fully developed coping mechanisms, making them especially vulnerable to adverse psychological outcomes [2]. In the wake of an earthquake, young survivors frequently exhibit posttraumatic stress symptoms (PTSS) and may develop full-blown posttraumatic stress disorder (PTSD), with prevalence rates reported to range from 2.5% to 60% in various studies [3]. The consequences of these conditions can be severe, affecting daily functioning, academic performance, and even brain development over the long term [4].

Recent research has increasingly focused on identifying the factors that contribute to the development of psychopathology following disasters [5]. While traditional vulnerability factors such as gender and pre-existing mental health conditions have long been recognized, more recent studies have emphasized the role of direct disaster exposure, including factors such as proximity to the epicenter, damage, and resource loss [6]. The well-documented “exposure effect” suggests that the closer individuals are to the epicenter, the higher the likelihood and severity of posttraumatic symptoms, with a substantial body of evidence supporting a dose–response relationship between exposure intensity and adverse outcomes [7,8,9,10].

In addition to individual exposure, familial factors play a critical role in determining post-disaster outcomes. Parental influence operates through multiple pathways, including emotional modeling, parenting behaviors, and stress transmission [11]. Children often internalize parental emotional responses, particularly during crises, where elevated parental distress can amplify the child’s own emotional reactions [12]. Parental PTSD and depression may also disrupt caregiving quality by reducing emotional availability and consistency—key elements in helping children process traumatic experiences [13,14,15]. In this way, parental mental health acts both as a direct influence on child outcomes and as a moderator of how children perceive and cope with external stressors. Moreover, a family’s socioeconomic status (SES) can exacerbate or buffer the impact of traumatic events [16]. Lower SES, often associated with financial burdens and limited access to resources, may intensify the stress experienced by families before, during, and after disasters. Conversely, strong social support networks—comprising both formal supports such as social services and informal supports from family and peers—can mitigate the negative effects of trauma [17,18]. However, even robust support systems may be overwhelmed in the context of extreme disaster exposure, reducing their effectiveness as protective factors [19]. This body of research highlights the importance of considering both direct exposure and the broader familial and social contexts when assessing post-disaster mental health.

To frame these interactions more explicitly, this study is conceptually informed by Bronfenbrenner’s ecological systems theory [20] and the transactional model of stress and coping [21]. Bronfenbrenner’s model [20] emphasizes how children’s development is shaped by nested environmental systems, ranging from immediate family to broader cultural and institutional structures. In the context of a disaster, parental psychological functioning belongs to the microsystem and directly interacts with the child’s own coping resources. The transactional model of stress and coping [21] further underscores the dynamic, reciprocal nature of this interaction, wherein the child’s response to trauma is shaped not only by the event itself but also by how caregivers appraise and respond to it. Together, these frameworks help clarify how parental distress and social support systems can either buffer or amplify trauma outcomes in youth.

Greece is a seismically active country with a history of significant earthquakes. For example, the 1999 earthquake in Athens resulted in substantial psychological distress among affected populations, particularly among children, who exhibited high rates of PTSS and depression compared to controls [22]. The current study focuses on the twin earthquake that struck Kefalonia in 2014—a high-magnitude, non-destructive event with relatively low community impact. Kefalonia, the largest of the Ionian Islands, is known for its high seismic activity and has a historical record of severe earthquakes, including a devastating 7.2 magnitude quake in 1953 [23]. By examining adolescents affected by the 2014 event, our study aims to extend the existing literature by investigating the prevalence and severity of PTSS, PTSD, and related emotional, behavioral, and sleep disturbances. Furthermore, we explore the roles of earthquake exposure, parental psychopathology, SES, and social support as predictors of these outcomes. In summary, this study seeks to provide a comprehensive understanding of how disaster exposure and familial factors interact to influence the mental health of young survivors. By doing so, it aims to inform the development of targeted, multi-layered interventions that address not only individual symptoms but also the broader familial and community dynamics essential for long-term recovery and resilience.

## 2. Materials and Methods

An earthquake sequence, comprising two main shocks occurring within a short time and spatial interval, struck Kefalonia Island, Greece, in early 2014. The first earthquake occurred on 26 January 2014, at 15:55, with a magnitude between 6.0 and 6.1 on the Richter scale. Although no fatalities or serious injuries were reported, the western part of Kefalonia—particularly the Paliki peninsula—suffered extensive damage to buildings and infrastructures, with the epicenter located in the south-eastern part of Paliki. Eight days later, on 3 February 2014, at 05:08, a second earthquake with a magnitude of approximately 5.9 to 6.0 struck the same area, exacerbating the damage caused by the first event. The epicenter of the second event was situated in the north-western part of Paliki peninsula. Despite recording the highest Peak Ground Acceleration (PGA) values ever documented in Greece, no fatalities or serious injuries were reported during this sequence.

In the aftermath of both earthquakes, severe damage was observed in technical constructions and infrastructure, historical monuments, public services, health units, nursing homes, schools, and residential buildings. For instance, about half of the houses in Lixouri were rendered uninhabitable, and several buildings in Argostoli sustained significant damage, while the road network was nearly destroyed. Additionally, the events triggered landslides, falling boulders, and soil liquefaction. In the port of Lixouri, the pier subsided after the first earthquake and collapsed following the second. Most injuries among residents were caused by falling objects and plaster, especially during the nighttime second earthquake [24].

For the purposes of this study, the affected areas were categorized based on their geographic proximity to the epicenters. The first area—Paliki province (Lixouri) [epicenter; high impact (HI)]—was the closest to the epicenters and experienced the most intense shaking (Mercalli intensities ranging from VII to VIII). The town of Argostoli [near epicenter; medium impact (MI))] is located 7–17 km from the epicenters and experienced shaking intensities between VI and VII. Districts situated 20–30 km from the epicenters [remote; low impact (LI)] experienced significantly less destruction, with shaking intensities lower than V.

### 2.1. Participants

The study sample comprised 520 students, aged 11–18 years, from schools in the three designated districts, with one parent of each student also included. This sample represented approximately 35% of the total student population on the island. Schools were randomly selected across all districts, ensuring equal representation of primary schools, gymnasiums, and high schools. Eligible participants were those who had experienced the twin earthquake eight months prior and were between 11 and 18 years old. Additional inclusion criteria included attendance at a mainstream school, proficiency in the Greek language (both oral and written), and the provision of informed consent by both students and their parents.

The final sample consisted of 502 adolescents and 474 parents from all three districts. Of the approximately 650 student–parent pairs invited to participate, 520 adolescents initially consented, resulting in an 80% adolescent response rate. However, 18 adolescent questionnaires were excluded due to missing or invalid responses, yielding a final analyzable sample of 502. Similarly, while all 520 families received parent questionnaires, only 474 were returned and included. In 28 cases, adolescent data were retained even though the corresponding parent forms were not submitted. For variables such as parental education, most data were derived from parent questionnaires; however, in cases where the parent questionnaire was missing or incomplete, adolescents provided information on parental education. These instances were used for descriptive statistics but excluded from analyses involving parent-reported clinical measures.

Data collection was conducted in October 2014, approximately eight months after the second earthquake event. Inclusion criteria for adolescents included (a) age between 11 and 18 years, (b) direct exposure to the twin earthquakes, (c) enrollment in a mainstream public school, and (d) fluency in written and spoken Greek. Parents were eligible if they were the legal guardian of a participating adolescent and agreed to complete the relevant questionnaires. Exclusion criteria included severe cognitive or communication impairments that could hinder reliable self-reporting. Each adolescent participant was matched with one corresponding parent, forming adolescent–parent dyads for analysis. In most cases (81.6%), the parent respondent was the mother. Recruitment yielded a demographically diverse sample, with representation across educational levels and socioeconomic strata. However, no data were collected on participants’ pre-earthquake psychiatric history, which represents a limitation in interpreting baseline vulnerability. Informed consent was obtained from parents via written forms distributed through the schools, while adolescents provided oral assent in the classroom before participation.

### 2.2. Procedure

After obtaining approval from the Ministry of Education, the research team contacted the principals of the selected schools to secure permission to conduct the study. Once approval was granted, classes were randomly selected from each participating school across the three educational levels—primary schools, gymnasiums, and high schools. Students received a letter outlining the study’s objectives and requesting both parental consent and their own assent. Parents completed the questionnaires at home, while students completed questionnaires in class in groups of 20–25 during a 40 min session. All questionnaires were completed anonymously and on a voluntary basis. Research assistants were present in the classrooms to explain the procedures and address any questions. The overall response rate was 80% for both students and parents.

### 2.3. Measures

All psychometric instruments used in this study were either officially validated Greek versions or had been previously used in Greek populations with established psychometric properties, as referenced below.

#### 2.3.1. Demographics

Parents’ and adolescents’ sociodemographic characteristics were assessed with an 18-item questionnaire. This self-reported questionnaire was developed by the investigators of the study and obtained sociodemographic data such as age, gender, socioeconomic status, and educational level.

#### 2.3.2. PTSS Symptoms

A 21-item self-report scale assessing PTSS symptom frequency over the previous month was administered to all students, utilizing the University of California at Los Angeles PTSD Reaction Index for DSM-IV, version 1 (UCLA-PTSD-RI) [25,26]. These items mirror the DSM-IV criteria for PTSD and rate the frequency of symptom occurrence on a 5-point Likert-type scale, ranging from none (0) to almost always (4). The UCLA PTSD-RI has shown good psychometric properties in various clinical and cultural settings, with a reported internal consistency ranging from 0.85 to 0.90 and excellent test-retest reliability [27]. Total scores of 38 are considered severe, 24 to 37 moderate, and 12 to 23 mild when used in children and adolescents. The internal consistency in this study was excellent (Cronbach’s α = 0.91).

The Impact of Event Scale-Revised (IES-R) [28] was used to measure parents’ severity of PTSS. The IES-R is a 22-item self-rating scale enabling the assessment of psychological trauma in three subscales (intrusion, avoidance, and hyperarousal). Parents were asked to rate their perceived stress response for each item on a 5-point Likert scale ranging from 0 = not at all to 4 = extremely according to the previous 7 days. Total scores for the IES-R are formed for all three subscales and range between 0 and 88, with higher scores reflecting a higher degree of PTSS. The Greek version of the IES-R had Cronbach’s alphas for the intrusion, avoidance, and hyperarousal scales of 0.72, 0.77, and 0.85, respectively [29]. In the present sample, the internal consistency was excellent (Cronbach’s α = 0.96). In the current study, the traumatic event was the parent’s experience of the twin earthquake.

#### 2.3.3. Earthquake Exposure

The Earthquake Exposure Questionnaire (EEQ) assessed the adolescents’ experiences and the degree of exposure to the earthquake. This 13-item ordinal self-report scale, developed by Kolaitis et al. [22], evaluates both objective and subjective aspects of exposure to a traumatic event. The EEQ inquires about experiences such as the death, injury, or disappearance of family members, damage to the home, and property loss. Each item is rated on a three-point scale, and the total score is used to classify exposure into three levels: no or mild exposure (score 0–3), moderate exposure (score 4–8), and severe exposure (score 9–13).

#### 2.3.4. Anxiety Symptoms

The Screen for Children’s Anxiety and Related Disorders–Short Form (SCARED-SF) [30] was used to evaluate the adolescents’ anxiety symptoms. This 5-item self-report scale is designed for children and adolescents aged 8 to 18 years and assesses anxiety disorder symptoms—such as panic/somatic complaints, generalized anxiety, separation anxiety, social anxiety, and school phobia—using a 3-point Likert scale (0 = almost never, 1 = sometimes, 2 = often). The current version comprises the five items from the original 41-item scale that demonstrated the highest loadings in a discriminant function analysis. Both the short and full versions of the SCARED have shown good psychometric properties [31]. The internal consistency in this sample was acceptable (Cronbach’s α = 0.64).

Parents and adolescents were invited to complete the state anxiety (S-Anxiety) scale of the State-Trait Anxiety Inventory (STAI)—form Y [32], a brief self-rating scale for the assessment of state and trait anxiety. State anxiety refers to the subjective and transitory feeling of tension, nervousness, and worry, which may be characterized by activation of the autonomous nervous system, at a given moment. The S-Anxiety scale consists of twenty statements that evaluate how the respondent feels “right now, at this moment”. In responding to the S-Anxiety scale, the subjects choose the number that best describes the intensity of their feelings: (1) not at all, (2) somewhat, (3) moderately, or (4) very much so. Scores on each subscale range from 20 to 80, with higher scores indicating greater state anxiety. The Greek version of the instrument has demonstrated excellent internal consistency [33]. The scale demonstrated good internal consistency for adolescents (Cronbach’s α = 0.87) and parents (Cronbach’s α = 0.86) in this study.

#### 2.3.5. Depressive Symptoms

Parental depressive symptoms were measured using the Beck Depression Inventory-II (BDI-II) [34]. The BDI-II consists of 21 items assessing the severity of depressive symptoms over the past two weeks. Each item presents four self-rated statements reflecting symptom severity, scored from 0 (absent) to 3 (most severe), resulting in a total score ranging from 0 to 63. According to the BDI-II manual, cutoff scores are defined as follows: no or minimal depression (<13), mild depression (14–19), moderate to severe depression (20–28), and severe depression (29–63). The scale’s good psychometric properties have been confirmed for the Greek population [35]. The internal consistency in this sample was very good (Cronbach’s α = 0.88).

#### 2.3.6. Sleep Problems

The brief version of the Athens Insomnia Scale (AIS-5) [36] was used to identify insomnia and sleep complaints in adolescents and their parents. The AIS-5 is a 5-item self-administered scale that measures insomnia severity based on the International Classification of Diseases, 10th Revision (ICD-10) insomnia criteria. These items assess difficulties in sleep initiation, awakenings during the night, early morning awakenings, total sleep time, and overall sleep quality, each rated on a 4-point Likert scale. Participants were asked to rate each item based on their experiences over the past month. The AIS-5 has demonstrated good psychometric properties [36]. The scale showed good internal consistency for both adolescents (Cronbach’s α = 0.713) and parents (Cronbach’s α = 0.743) in this study.

#### 2.3.7. Social Support

To assess the level of social support for both adolescents and their parents, the Oslo 3 Social Support Scale (OSS-3) [37] was used. The OSS-3 is a 3-item self-administered rating scale, with Item 1 rated on a 4-point scale and Items 2 and 3 on 5-point scales. Respondents indicate their perception of social support from family, friends, and neighbors. Total scores range from 3 to 14, with scores of 3–8 indicating poor support, 9–11 moderate support, and 12–14 strong support. The instrument has been applied in several large-scale population-based surveys in different settings, e.g., the European KIDSCREEN Study [38]. Internal consistency in the present study was acceptable in both adolescents (Cronbach’s α = 0.65) and parents (Cronbach’s α = 0.75).

#### 2.3.8. Emotional/Behavioral Problems

The Strengths and Difficulties Questionnaire (SDQ) [39] was administered to parents to evaluate their children’s adjustment difficulties and prosocial behavior. The SDQ consists of 25 items organized into five subscales: emotional problems, conduct problems, hyperactivity, peer problems, and prosocial behavior. Each item is rated on a 3-point Likert scale (0 = not true, 1 = somewhat true, 2 = certainly true). For negatively worded items, scores are assigned as rated, while positively worded items are reverse-scored (2–0). A total difficulty score is computed by summing the scores of the first four subscales, yielding a range from 0 to 40, with higher scores indicating more adjustment difficulties. The instrument’s factor structure and psychometric properties have been documented in a representative nationwide sample of Greek adolescents [40]. In this study, the measure showed good internal consistency for both adolescents (Cronbach’s α = 0.81) and parents (Cronbach’s α = 0.80).

### 2.4. Statistical Analysis

Descriptive analyses were conducted for both continuous and categorical variables. Means and standard deviations were calculated for continuous variables, while categorical variables were presented as absolute and relative frequencies. Participants were divided into three groups based on their geographic proximity to the epicenter (i.e., on epicenter, near epicenter, and remote). Bivariate associations between demographic characteristics and outcome measures were assessed using Chi-square tests. Pearson’s correlation coefficients and Spearman rank correlation analyses were performed for each outcome variable. Multiple hierarchical linear regression analyses were conducted to examine the associations between potential risk factors and the severity of clinical outcomes in children. Gender, age, socioeconomic status, exposure index, social support, and parental clinical characteristics (PTSD, depression, anxiety, and sleep problems) were entered as independent variables, with UCLA-PTSD, SDQ, SCARED, and Insomnia scores as dependent variables. Standardized regression coefficients (β), F statistics, R^2^, and changes in R^2^ (ΔR^2^) for each step were calculated. Two-tailed *p*-values < 0.05 were considered statistically significant. Data were analyzed using SPSS version 28.0 (SPSS Inc., Chicago, IL, USA).

## 3. Results

### 3.1. Demographic and Clinical Characteristics

The demographic information and clinical characteristics for the study sample are summarized in Table 1. Among the 502 adolescent participants, 39.6% (N = 199) were male with a mean age of 14.04 years (SD = 2.36), and 60.4% (N = 303) were female with a mean age of 14.32 years (SD = 2.43). Overall, 60.2% (N = 293) of adolescent participants reported moderate exposure to the earthquake, and 3.1% (N = 15) reported high exposure. Regarding socioeconomic status, 25.4% (N = 123) of adolescents were in the lower SES category, 55.8% (N = 270) in middle SES, and 18.8% (N = 91) in upper SES, with no significant differences between genders.

The UCLA-PTSD scores for adolescents ranged from 0 to 59, with a mean of 14.71 (SD = 11.51); 5.2% (N = 26) had scores suggestive of probable PTSD (UCLA-PTSD > 38). Compared to boys, girls had significantly higher total UCLA-PTSD scores [t(500) = 2.84, *p* < 0.01], STAI scores [t(463) = 3.66, *p* < 0.01], SCARED scores [t(488) = 5.01, *p* < 0.01], and Insomnia scores [t(459) = 4.57, *p* < 0.01]. In terms of proximity to the epicenter, significant differences were found in UCLA-PTSD scores [F(2, 499) = 4.00, *p* = 0.02] and STAI scores [F(2, 462) = 3.51, *p* = 0.03]. Bonferroni post hoc analyses revealed that adolescents in the epicenter group exhibited significantly higher PTSD symptoms compared to those from remote regions.

For parents, the mean age was 44.39 years (SD = 6.32), with the majority being mothers (81.6%; N = 409). Half of the parents (51.40%, N = 258) reported that their highest educational attainment was high school/lyceum. The mean IES-R score was 30.83 (SD = 22.88), and the prevalence of probable PTSD (IES-R > 33) among parents was 44.3% (N = 197). Parents in the epicenter group had significantly higher IES-R scores [F(2, 442) = 18.80, *p* < 0.001], STAI scores [F(2, 487) = 6.68, *p* < 0.01], OSLO scores [F(2, 491) = 3.05, *p* < 0.05], and Insomnia scores [F(2, 475) = 8.79, *p* < 0.001]. Significant differences were also observed in the prevalence of probable PTSD among parents, with rates of 57.4% (N = 113) in the epicenter group, 26.4% (N = 52) in the near-epicenter group, and 16.2% (N = 32) in the remote group [χ^2^(2, 445) = 29.78, *p* < 0.01]. Bonferroni post hoc analyses further revealed that parents from the epicenter exhibited significantly higher IES-R and STAI scores than those from near-epicenter and remote regions.

### 3.2. Correlation Analyses

The correlations between the clinical characteristics of parents and adolescent’s outcome scores are presented in Table 2. With the exception of adolescent and parental social support, all outcome measures—both within and between groups—were significantly correlated. In adolescents, emotional/behavioral symptoms showed the highest correlation with PTSS (r = 0.63, *p* < 0.001), followed by sleep problems and anxiety symptoms. Similar patterns were observed among parents, where sleep problems exhibited the strongest correlation with PTSS (r = 0.55, *p* < 0.001), followed by anxiety and depressive symptoms. In addition, PTSS levels were positively associated with earthquake exposure in both adolescents (r = 0.33, *p* < 0.001) and parents (r = 0.39, *p* < 0.001).

### 3.3. Prediction of PTSS, Emotional/Behavioral, Anxiety Symptoms, and Sleep Problems

Hierarchical regression analyses were conducted, with PTSS, emotional/behavioral problems, anxiety, and sleep problems as dependent variables (Table 3). In step 1, demographic characteristics were entered as predictors, accounting for 3.9% of the variance in PTSS (ΔR^2^ = 0.039, *p* < 0.01), 6% of the variance in emotional/behavioral problems, 5.9% of the variance in anxiety symptoms (ΔR^2^ = 0.059, *p* < 0.01), and 12.5% of the variance in sleep problems (ΔR^2^ = 0.125, *p* < 0.01).

In step 2, earthquake-related predictors—specifically, earthquake exposure and social support—were added, resulting in significant increases in the cumulative variance explained: an additional 11.5% for PTSS (ΔR^2^ = 0.115, *p* < 0.01), 16.2% for emotional/behavioral problems (ΔR^2^ = 0.162, *p* < 0.01), 4.8% for anxiety symptoms (ΔR^2^ = 0.048, *p* < 0.01), and 13% for sleep problems (ΔR^2^ = 0.013, *p* < 0.01).

In step 3, parental clinical characteristics—namely, parental PTSS, depressive symptoms, anxiety symptoms, and sleep problems—were entered. These variables accounted for an additional 9.3% of the variance in PTSS (ΔR^2^ = 0.093, *p* < 0.01), 5.3% of the variance in emotional/behavioral problems (ΔR^2^ = 0.053, *p* < 0.01), 3.8% of the variance in anxiety symptoms (ΔR^2^ = 0.038, *p* < 0.01), and 6.6% of the variance in sleep problems (ΔR^2^ = 0.066, *p* < 0.01).

In conclusion, the final models explained 22.4% of the variance in PTSS, 25.5% in emotional/behavioral problems, 14% in anxiety symptoms, and 30.0% in sleep problems. These results identified earthquake-related factors as the strongest predictors for all outcome variables.

## 4. Discussion

This study investigated the prevalence and severity of posttraumatic stress symptoms (PTSS) and associated emotional, behavioral, anxiety, and sleep disturbances in children and adolescents exposed to the twin earthquake in Kefalonia, Greece. Additionally, we examined the roles of earthquake exposure, parental psychopathology, socioeconomic status (SES), and social support in predicting these outcomes. The findings offer important insights into the multifaceted nature of trauma responses in youth and underscore the significance of family and community contexts in post-disaster mental health.

Prevalence rates from our sample support these patterns: 5.2% of adolescents exhibited probable PTSD, with girls reporting significantly higher levels of symptoms. Among parents, 44.3% met criteria for probable PTSD, with the highest prevalence (57.4%) observed in the epicenter group. Earthquake exposure was strongly associated with increased PTSS, anxiety, and sleep disturbances in both adolescents and parents, while parental psychopathology—especially PTSS and depression—was closely linked to elevated symptom severity in children.

Our results confirm that degree of earthquake exposure is a significant predictor of adverse psychological outcomes [6,7,8,10]. Adolescents residing in areas closest to the epicenters exhibited higher levels of PTSS and related symptoms compared to those in more remote regions. This finding is consistent with the dose–response relationship observed in previous research and highlights that even a non-destructive yet high-magnitude earthquake can have a profound psychological impact [6,7,8,10]. Importantly, our study demonstrates that substantial mental health challenges can arise, even when physical injuries and fatalities are minimal, emphasizing the need for mental health interventions, regardless of the level of physical destruction.

The prevalence of probable PTSD in adolescents in our sample falls at the lower end of the range reported in international studies, where post-earthquake PTSD rates in children have been found to vary widely—from 2.5% to as high as 60%, depending on the context and methodology [3]. One possible explanation for the relatively lower prevalence observed in our Greek sample is the population’s high level of familiarity with seismic activity, particularly in historically earthquake-prone regions such as Kefalonia. Widespread public awareness, school-based preparedness drills, and strict construction codes may contribute to a sense of environmental mastery and psychological resilience. Nevertheless, the high levels of PTSD among parents underscore that, even in well-prepared communities, the psychological burden may shift across family subsystems, particularly through mechanisms of stress contagion and caregiving disruption [11].

Parental factors emerged as critical determinants of children’s outcomes. Our hierarchical regression analyses revealed that parental PTSS, depressive, and anxiety symptoms significantly contributed to the variance in adolescents’ PTSS, emotional/behavioral problems, anxiety, and sleep disturbances. These findings corroborate earlier studies documenting the intergenerational transmission of trauma [14,15]. The data suggest that the psychological well-being of parents not only affects their own functioning but also shapes the recovery trajectories of their children. Several mechanisms may underlie this relationship. Parental distress can contribute to emotional contagion, whereby children absorb and mirror parental fear and dysregulation. It may also impair parents’ ability to provide emotional attunement and model adaptive coping behaviors. Heightened psychopathology may further destabilize family systems, diminish children’s sense of safety, and disrupt the routines that support emotional regulation. While our study design did not permit formal mediation or path analysis, future research should examine these pathways empirically using longitudinal and structural models. As such, post-disaster interventions should incorporate integrated, family-centered approaches that target both parental distress and child symptomatology.

Social support also played a protective role, as higher levels of perceived support were associated with lower levels of PTSS and other adverse outcomes in both adolescents and parents. This finding aligns with previous research indicating that robust social networks and the availability of formal support services can buffer the negative effects of trauma [17,18]. However, our results further suggest that even in the presence of social support, high levels of earthquake exposure can disrupt these buffering mechanisms. This may result from the overwhelming nature of the disaster, which not only strains individual coping resources but may also lead to a breakdown in communal and familial support systems [19].

Gender differences in the manifestation of trauma symptoms were evident, with girls reporting significantly higher levels of PTSS, anxiety, and sleep problems compared to boys. These findings are in line with existing literature suggesting that female adolescents may be more vulnerable to the psychological effects of trauma [6]. This gender difference may be attributed to several possible mechanisms. Biologically, females may exhibit heightened hypothalamic–pituitary–adrenal (HPA) axis reactivity to stress, increasing vulnerability to trauma-related symptoms [41]. Psychologically, girls often engage more in emotion-focused coping and rumination, which have been linked to elevated PTSS [42]. Socioculturally, girls may be more likely to express distress openly, or to perceive interpersonal threats more acutely [43]. These differences in coping styles and stress appraisal may help explain the elevated symptom levels observed in female adolescents following disaster exposure. This gender disparity underscores the importance of tailoring post-disaster interventions to address the specific needs of different groups, ensuring that strategies are sensitive to gender-related differences in trauma response.

The role of SES in our study was less pronounced than that of direct earthquake exposure and familial factors. Although SES was included as a predictor in our regression models, its impact was relatively modest. This may be attributable to the overall homogeneity of the sample or to the possibility that the direct effects of trauma and family dynamics overshadowed SES differences in this context. Future research should further explore the conditions under which SES might have a more substantial impact on disaster-related psychopathology.

From a clinical standpoint, these results highlight the urgent need for multi-layered intervention programs in post-disaster contexts. Routine mental health screenings should be implemented for both adolescents and their parents following seismic events, even in the absence of physical injuries. Interventions should be trauma-informed and incorporate family-based therapeutic strategies to address the bidirectional emotional influences between parent and child. Particular attention should be paid to parental PTSD and depression, as addressing these issues can indirectly alleviate children’s symptoms. Additionally, mental health services should be tailored to the local cultural context and include school-based support for early detection and intervention.

Our study was conducted eight months after the earthquake, a timing that may capture a transitional phase in recovery. This post-disaster period is critical for early intervention, as timely support can leverage the high neuroplasticity of children and adolescents to promote faster recovery and potentially prevent the persistence of symptoms. Longitudinal studies are needed to track the evolution of these symptoms over time and to determine the long-term efficacy of various intervention strategies.

Several methodological strengths bolster the validity of our findings. The study employed a large, representative sample, which accounted for approximately 35% of the island’s student population. Furthermore, the use of well-validated instruments that were appropriately translated and pretested in the local context enhanced the reliability of our measures.

In summary, our study emphasizes that both the intensity of earthquake exposure and familial factors—particularly parental psychopathology—significantly contribute to the development of PTSS and comorbid problems in adolescents. These findings have important clinical and public health implications. They suggest that effective post-disaster interventions should adopt a multi-layered approach, combining individual therapy with family-centered and community-based support. By addressing both the immediate psychological needs of children and the broader familial context, such interventions may not only ameliorate acute distress but also foster long-term resilience and recovery in disaster-affected populations.

Notably, the mechanisms observed in this study—particularly the influence of parental distress and social support on youth outcomes—remain relevant in the context of more recent disasters, such as the 2025 Southern California wildfires and the Myanmar earthquake of the same year. While detailed psychological data from these events are not yet available, the findings of our study underscore enduring patterns of post-disaster vulnerability and resilience that may help guide mental health response efforts in both current and future contexts.

### 4.1. Limitations

Several limitations should be acknowledged. First, the cross-sectional design restricted our ability to infer causal relationships between earthquake exposure, parental psychopathology, and child outcomes. Second, the reliance on self-report measures introduced the potential for response bias, including social desirability or inaccurate recall. Third, the study was limited to adolescents attending mainstream public schools and to Greek-speaking participants, potentially excluding vulnerable subpopulations such as children with disabilities, recent migrants, or those in alternative educational settings. Fourth, while validated Greek versions of the instruments were used, shared method variance due to using self-reports across informants may have inflated associations. Lastly, the internal consistency of the SCARED-SF in this sample was relatively low (Cronbach’s α = 0.64), potentially limiting the reliability of anxiety symptom assessment.

Although data collection took place in 2014, the publication of results was delayed due to logistical and institutional constraints, including disruptions during the COVID-19 pandemic. The current manuscript reflects a reanalysis of the dataset and incorporates recent conceptual advances in trauma research. While the data were collected in 2014, they may offer enduring insights into disaster-related trauma that remain relevant to current global events, especially in light of ongoing ecological and humanitarian crises.

### 4.2. Future Directions

Future research should consider longitudinal designs to examine how trauma symptoms evolve over time and identify which children recover naturally versus those at risk for chronic difficulties. Studies involving more diverse populations—including linguistic minorities, private school students, and younger children—would enhance generalizability. Further investigation is also needed into the specific mechanisms linking parental and child mental health, including parenting behaviors, attachment quality, and family coping styles. Finally, intervention studies testing family-based, school-based, and community-level supports would help translate the findings into evidence-based post-disaster practices.

## 5. Conclusions

This study highlights the multifactorial nature of psychological distress in youth following natural disasters, emphasizing the combined influence of earthquake exposure, parental psychopathology, and social support. Even in a relatively low-fatality context such as the 2014 Kefalonia earthquakes, adolescents exhibited significant rates of posttraumatic stress and comorbid symptoms, with parental mental health emerging as a key predictor. Our findings reinforce the importance of adopting a systemic, family-centered approach to post-disaster mental health interventions.

Notably, the mechanisms observed in this study—particularly the influence of parental distress and social support—remain highly relevant in current disaster contexts. Although psychological data from these events are still emerging, our results underscore enduring patterns of vulnerability and resilience that can inform early intervention and preparedness strategies across diverse cultural and geographic settings.

Effective disaster response must integrate individual, familial, and community-level supports to address not only immediate distress but also to promote long-term adaptation and recovery. Continued research and policy attention are needed to ensure that mental health responses are timely, context-sensitive, and inclusive of both children and their caregiving environments.

## Figures and Tables

**Table 1 healthcare-13-01249-t001:** Demographic and clinical characteristics of adolescents and their parents, stratified by proximity to the earthquake epicenter (N = 502).

Variables		Total (N = 502)	Near Epicenter (N = 167)	On Epicenter (N = 207)	Remote (N = 128)	χ^2^/F Value ^a^
**Adolescents**						
Age (years)	M ± SD	14.21 (2.40)	13.55 (2.51)	14.35 (2.29)	14.84 (2.25)	11.28 *
Gender	N (%)					
Boy		199 (39.6)	67 (33.7)	88 (44.2)	44 (22.1)	2.21
Girl		303 (60.4)	100 (33.0)	119 (39.3)	84 (27.7)	
School level	N (%)					54.83 *
Elementary		167 (33.3)	89 (53.3)	56 (33.5)	22 (13.2)	
Gymnasium		147 (29.3)	26 (17.7)	77 (52.4)	44 (29.9)	
Lyceum		188 (37.5)	52 (27.7)	74 (39.4)	63 (34.4)	
EEQ	N (%)					62.29 *
Low		179 (36.8)	73 (40.8)	36 (20.1)	70 (39.1)	
Medium		293 (60.2)	89 (30.4)	150 (51.2)	54 (18.4)	
High		15 (3.1)	2 (13.3)	13 (86.7)	0 (0)	
UCLA-PTSD-RI	M ± SD	14.71 (11.51)	15.47 (12.75)	15.63 (10.74)	12.50 (10.70)	4.00 **
UCLA-PTSD-RI > 38	N (%)	26 (5.2)	13 (50.0)	10 (38.5)	3 (11.3)	4.03
SDQ	M ± SD	8.91 (5.89)	8.45 (5.90)	9.09 (5.61)	9.21 (6.29)	0.25
S-Anxiety	M ± SD	38.90 (10.33)	37.73 (11.01)	40.39 (10.22)	37.96 (9.34)	3.31 *
OSSS-3	M ± SD	11.48 (2.06)	11.44 (1.92)	11.44 (2.16)	11.58 (2.09)	0.19
SCARED	M ± SD	1.69 (1.85)	1.81 (1.93)	1.71 (1.84)	1.50 (1.73)	1.00
AIS-5	M ± SD	14.39 (4.78)	14.75 (4.90)	14.45 (4.61)	13.79 (4.78)	1.34
**Parents**						
Father, N (%)		92 (18.4)	31 (6.2)	38 (7.6)	23 (4.8)	0.19
Mother, N (%)		409 (81.6)	136 (27.1)	168 (33.5)	105 (21.0)	
Age (years)	M ± SD	44.39 (6.32)	44.19 (6.35)	44.37 (6.02)	44.70 (6.78)	0.21
EEQ	N (%)					29.78 *
Low		120 (37.9)	55 (45.8)	11 (9.2)	54 (45.0)	
Medium		174 (54.9)	47 (27.0)	77 (44.3)	50 (28.7)	
High		23 (7.3)	4 (17.4)	19 (82.6)	0 (0)	
Parent’s Education level	N (%)					25.21 *
Low		64 (12.8)	8 (12.5)	36 (56.3)	20 (31.3)	
Medium		258 (51.6)	79 (30.6)	115 (44.6)	64 (24.8)	
High		178 (35.6)	79 (44.4)	55 (30.9)	44 (24.7)	
SES	N (%)					4.84
Low		123 (25.4)	37 (30.1)	60 (48.8)	27 (21.1)	
Medium		270 (55.8)	95 (35.2)	108 (40.0)	68 (24.8)	
High		91 (18.8)	32 (35.2)	32 (35.2)	27 (29.7)	
IES-R	M ± SD	30.83 (22.88)	26.72 (22.55)	38.04 (22.06)	23.46 (21.18)	18.80 *
IES-R ≥ 33	N (%)	197 (44.3)	52 (26.4)	113 (57.4)	32 (16.2)	29.78 *
BDI-II	M ± SD	6.66 (6.87)	6.06 (6.84)	7.27 (6.80)	6.59 (6.94)	1.43
S-Anxiety	M ± SD	43.50 (10.66)	41.87 (10.07)	45.56 (11.52)	42.62 (9.54)	6.68 *
OSSS-3	M ± SD	10.97 (2.42)	10.61 (2.65)	11.22 (2.36)	11.04 (2.13)	3.05 *
AIS-5	M ± SD	15.95 (5.42)	15.04 (5.30)	17.19 (5.50)	15.20 (5.10)	8.79 *

^a^ Bonferroni adjusted; * *p* < 0.001; ** *p* < 0.05.

**Table 2 healthcare-13-01249-t002:** Correlation matrix of key outcome variables in adolescents and parents (N = 502).

	1	2	3	4	5	6	7	8	9	10	11	12
**Adolescents**												
1. EEQ	1											
2. UCLA-PTSD-RI	0.33 ***	1										
3. SDQ	0.25 ***	0.63 ***	1									
4. S-Anxiety	0.34 ***	0.53 ***	0.57 ***	1								
5. OSSS-3	−0.09	−0.23 ***	−0.40 ***	−0.28 ***	1							
6. SCARED-SF	0.16 ***	0.55 ***	0.46 ***	0.47 ***	−0.22 ***	1						
7. AIS-5	0.31 ***	0.59 ***	0.44 ***	0.44 ***	−0.30 ***	0.44 ***	1					
**Parents**												
8. EEQ	0.48 ***	0.16 **	0.12 *	0.20 *	−0.10	0.20 ***	0.20 ***	1				
9. IES-R	0.26 ***	0.34 ***	0.21 ***	0.26 ***	−0.06	0.22 ***	0.24 ***	0.39 ***	1			
10. BDI-II	0.19 ***	0.30 ***	0.31 ***	0.18 ***	−0.13 **	0.23 ***	0.27 ***	0.19 ***	0.45 ***	1		
11. OSSS-3	−0.03	−0.11 *	−0.14	−0.06	0.25 ***	−0.11 **	−0.08	−0.10	−0.20 ***	−0.35 ***	1	
12. AIS-5	0.22 ***	0.30 ***	0.13 **	0.24 ***	−0.10 *	0.21 ***	0.31 ***	0.31 ***	0.55 ***	0.44 ***	−0.24 ***	1
13. S-Anxiety	0.29 ***	0.24 ***	0.22 ***	0.28 ***	−0.04	0.19 ***	0.22 ***	0.25 ***	0.49 ***	0.59 ***	−0.25 ***	0.49 ***

Note: PTSS = Posttraumatic stress symptoms; * *p* < 0.05; ** *p* < 0.01; *** *p* < 0.001.

**Table 3 healthcare-13-01249-t003:** Hierarchical regression analyses predicting adolescent PTSS, emotional/behavioral problems, anxiety, and sleep problems.

	UCLA-PTSD-RI			SDQ			SCARED-SF			AIS-5		
	Step 1 (β)	Step 2 (β)	Step 3 (β)	Step 1 (β)	Step 2 (β)	Step 3 (β)	Step 1 (β)	Step 2 (β)	Step 3 (β)	Step 1 (β)	Step 2 (β)	Step 3 (β)
**Step 1 Demographic characteristics**												
Age	0.594 *			0.491 **			0.033			0.531 ***		
Gender	−3.029 *			−1.022			−0.873 ***			−1.970 ***		
SES_1 Medium	−0.224			−0.921			−0.217			−0.333		
SES_2	1.936			−0.690			−0.021			0.701		
**Step 2 Earthquake related predictors**												
EEQ		1.770 ***			0.557 ***			0.096 *			0.584 ***	
OSSS-3		−0.994 ***			−1.008 ***			−0.173 ***			−0.645 ***	
**Step 3 Parental clinical characteristics**												
IES-R			0.850 **			0.020			0.008			0.003
BDI-II			0.288 **			0.190 ***			0.027			0.055
S-Anxiety			−0.106			0.017			0.007			0.001
AIS-5			0.282 **			−0.112			0.028			0.192 ***
F	3.785 **	11.375 ***	12.292 ***	5.872 ***	17.736 ***	14.014 ***	5.996 ***	7.511 ***	7.049 ***	12.660 ***	19.931 ***	16.343 ***
R^2^	0.039	0.150	0.243	0.059	0.221	0.274	0.060	0.108	0.161	0.125	0.254	0.320
Adjusted R^2^	0.028	0.137	0.224	0.049	0.209	0.255	0.050	0.094	0.138	0.115	0.241	0.300
△R^2^	0.039 **	0.115 ***	0.091 ***	0.059 ***	0.162 ***	0.053 ***	0.060 ***	0.048 ***	0.053 ***	0.125 ***	0.128 ***	0.066 ***

Note: Education_1: Medium vs. low SES; Education_2: High vs. low SES; * *p* < 0.05; ** *p* < 0.01; *** *p* < 0.001.

## Data Availability

The datasets used and/or analyzed during the current study are available from the corresponding author on reasonable request.

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
