# Peer review of "Unraveling Youth Trauma and Parental Influence After Twin Earthquakes"

_healthcare, 2025, doi:10.3390/healthcare13111249_

Round 1

Reviewer 1 Report

Comments and Suggestions for Authors

Dear authors,

First of all, I would like to thank you for the invitation to review your article "Unraveling Youth Trauma and Parental Influence After Twin Earthquakes." I would also like to congratulate you, as I find it a very well-developed article. However, I believe it could be relevant to make some improvements:

Introduction
It would be advisable to improve the explanation and conceptualization of parental influence.

Materials and Methods
The explanation of the analyzed areas is very well developed.
When was the sample collected?
What were the inclusion and exclusion criteria for the sample collection?
How was informed consent and assent obtained from the participants?
The scales seem very well chosen; however, I could not find evidence as to whether the Greek versions were used or if these scales are validated for that context.
It should be explicitly stated which ethics committee approved the conduct of this research.

Results
Please adjust Tables 1, 2, and 3, as the tables are misaligned and cannot be properly viewed or read.
The results are well developed; only the formatting of the tables needs to be improved.

Discussion
I believe it is advisable, at the beginning of the discussion, to state the prevalences found so that it serves as a summary of the results.
It is very important to clearly state the clinical implications of these findings.
It is necessary to relate the results to other international research. Ideally, compare with prevalences from other places and discuss why the results are so particular in Greece. Perhaps the population is accustomed to earthquakes, as in other countries, and maybe the prevalences are not as high...
Please, in the discussion, explicitly include a section on "limitations" and another on "future directions of the research."

Author Response

Reviewer Comment 1:

It would be advisable to improve the explanation and conceptualization of parental influence.

Response to Reviewer: We appreciate the reviewer’s suggestion to strengthen the conceptualization of parental influence in the Introduction. We revised the relevant paragraph to elaborate on the mechanisms by which parental mental health affects child outcomes, including emotional modeling, disrupted caregiving, and stress transmission. This addition clarifies the multidimensional role of parental influence in post-disaster adjustment and is now included in the Introduction.

Reviewer Comment 2:

When was the sample collected?
What were the inclusion and exclusion criteria for the sample collection?
How was informed consent and assent obtained from the participants?

Response to Reviewer:

We appreciate the reviewer’s request for clarification regarding the sample collection process. We have now specified in the Participants section that data were collected in October 2014, eight months after the earthquake. We also added detailed inclusion and exclusion criteria for adolescents and parents, and described the procedures for obtaining written parental consent and oral adolescent assent.

Reviewer Comment 3:

The scales seem very well chosen; however, I could not find evidence as to whether the Greek versions were used or if these scales are validated for that context.
It should be explicitly stated which ethics committee approved the conduct of this research.

Response to Reviewer:

Thank you for this valuable observation. We have clarified in the Measures section that all instruments used were validated Greek versions or had prior documented use in Greek populations with acceptable psychometric properties. Furthermore, we revised the Institutional Review Board Statement to explicitly state that ethics approval was granted by the Ethics Committee of the Ministry of Education of the Hellenic Republic.

Reviewer Comment 4:

Please adjust Tables 1, 2, and 3, as the tables are misaligned and cannot be properly viewed or read.

Response to Reviewer:

We thank the reviewer for pointing out the formatting issues with Tables 1, 2, and 3. We have carefully revised these tables to correct misalignments and improve readability. All tables now use consistent formatting with clearly defined columns, aligned values, and standardized footnotes.

Reviewer Comment 5:

I believe it is advisable, at the beginning of the discussion, to state the prevalences found so that it serves as a summary of the results.

Response to Reviewer:

Thank you for your helpful suggestion. In response, we have added a new paragraph after the first paragraph of the Discussion that summarizes the key prevalence rates and their distribution across subgroups. This addition serves as a concise results recap and reinforces the importance of the study’s findings.

Reviewer Comment 6:

It is very important to clearly state the clinical implications of these findings.

Response to Reviewer:

Thank you for your insightful suggestion. We have added a new paragraph toward the end of the Discussion, just before the section on timing and longitudinal follow-up, to clearly articulate the clinical implications of our findings. This placement allows the reader to reflect on the applied relevance of the results after the full discussion of predictors and contextual factors.

Reviewer Comment 7:

It is necessary to relate the results to other international research. Ideally, compare with prevalences from other places and discuss why the results are so particular in Greece. Perhaps the population is accustomed to earthquakes, as in other countries, and maybe the prevalences are not as high...

Response to Reviewer:

We appreciate the reviewer’s suggestion to contextualize our findings internationally. In response, we have added a paragraph comparing our prevalence data to PTSD rates reported in other countries following earthquakes. We also discuss cultural familiarity with seismic activity in Greece as a potential factor contributing to the relatively lower adolescent PTSD rates, while acknowledging the continued vulnerability in families, especially among parents.

Reviewer Comment 8:

Please, in the discussion, explicitly include a section on "limitations" and another on "future directions of the research."

Response to Reviewer:

We thank the reviewer for this important suggestion. In response, we added two clearly labeled subsections—Limitations and Future Directions—to the end of the Discussion. These sections address the study’s methodological constraints and propose concrete avenues for future research, including longitudinal tracking, inclusion of diverse populations, and intervention-focused studies.

Reviewer 2 Report

Comments and Suggestions for Authors

Author Response

Reviewer Comment 1:

The background effectively establishes the link between earthquake exposure and PTSS, it would benefit from a more robust theoretical grounding. The authors might consider integrating established models such as Bronfenbrenner’s ecological systems theory or the transactional model of stress and coping to frame the bidirectional influence between child and parental psychological functioning. This would provide readers with a clearer conceptual basis for understanding how familial and contextual variables interact following trauma exposure.

Response to Reviewer:

We thank the reviewer for the excellent suggestion to strengthen the theoretical foundation of the manuscript. In response, we have incorporated a new paragraph in the Introduction referencing Bronfenbrenner’s ecological systems theory and the transactional model of stress and coping. These frameworks provide a more robust conceptual basis for understanding the bidirectional and context-dependent influences between parental and child psychological responses after trauma exposure.

Reviewer Comment 2:

The manuscript references the use of “standardized self-report measures,” but it lacks specificity. It is essential to name each instrument used (e.g., UCLA PTSD Reaction Index, GAD-7, PSQI), indicate the domains they assess, and briefly comment on their psychometric properties, including internal consistency (Cronbach’s alpha) within this sample. Information on whether the instruments were validated for use with Greek-speaking populations is also important, especially when dealing with cross-cultural generalizability.

Response to Reviewer:

We appreciate the reviewer’s recommendation to enhance the description of measurement tools and their psychometric properties. In response, we revised Section 2.3 (Measures) to (a) briefly describe the psychological domains assessed by each instrument, (b) report Cronbach’s alpha values for internal consistency where available, and (c) reaffirm the use of validated Greek versions or prior applications in Greek samples. These additions improve transparency and strengthen the methodological rigor of the study.

Reviewer Comment 3:

The manuscript mentions the participation of 502 adolescents and 474 parents, but it is not clear how these participants were recruited, whether parental-child dyads were matched, or what the demographic composition (e.g., socioeconomic status, education levels, prior mental health history) was. This information is crucial for assessing the generalizability and representativeness of the sample.

Response to Reviewer:

We thank the reviewer for this important observation. To clarify the sample structure, we revised Section 2.1 (Participants) to explicitly state that adolescent–parent dyads were used, and we added a sentence summarizing key demographic data. We also acknowledged the absence of data on prior mental health history as a limitation in assessing baseline vulnerability.

Reviewer Comment 4:

The finding that girls reported significantly higher PTSS than boys is consistent with broader literature, but this should be contextualized with relevant citations. The discussion could explore potential biological, psychological, and sociocultural explanations for this gender disparity, such as differential socialization processes or coping mechanisms.

Response to Reviewer:

We appreciate the reviewer’s suggestion to elaborate on the observed gender differences in trauma outcomes. We have expanded the relevant section in the Discussion to include potential biological, psychological, and sociocultural mechanisms—such as stress reactivity, coping strategies, and gendered socialization—that may account for higher PTSS in girls. Relevant literature has been cited to support this interpretation.

Reviewer Comment 5:

The conclusion that parental psychopathology and reduced social support exacerbate youth trauma responses is compelling but requires more discussion. The manuscript should examine possible mechanisms such as emotional contagion, parental modeling of maladaptive coping, or increased familial stress that could mediate this relationship. Including a path analysis or mediation model, if data permits, could strengthen this contribution.

Response to Reviewer:

We thank the reviewer for this valuable suggestion. In the Discussion, we have elaborated on potential mechanisms linking parental psychopathology and youth trauma symptoms, including emotional contagion, maladaptive coping modeling, and increased familial stress. While our cross-sectional design precluded path or mediation analyses, we acknowledge this as a direction for future research and have explicitly stated this in the revised manuscript.

Reviewer 3 Report

Comments and Suggestions for Authors

1.  The calculation of the response rate needs clarification.  Using 520 as those who responded, an 80% response rate would mean that the original sample was 650 (?).  However, it is not clear how the sample analyzed appeared to be 502.  What happened to the other 18?  Did the final sample include both an adolescent and a parent?  Were there situations where one but not both responded?  It also seems unusual that the response rate for both parents and adolescents was 80% when the study presumably began with the same number of potential respondents/families and yet the final sample was 502 adolescents and 474 parents.  For example, on parental education, if that was obtained from the parents, the numbers add up to 500, more than 474.  Did that mean that parental education was reported by the adolescents or was that data available from other sources, like the schools?  These issues are probably not major but they should be cleared up.

2.  The study would seem to have been approved in October 2014 but now it's April 2025.  Any particular reason for the long delay in preparing these results?  Furthermore, the earthquakes occurred in February 2014 and the study was on page 10 said to have been conducted six months after the earthquakes but the approval wasn't until October, which is eight to nine months after the earthquakes.  Was approval obtained a couple of months after the study was begun?  Not necessarily a big issue but something that needs clarification. 

3.  As an historical side note, the USA ship, the USS Salem, was a heavy cruiser that helped the effected communities from the 1953 earthquake with food, water, and electricity.  That ship survives, unlike others that have been scrapped, as a museum ship in the Boston, USA area and its bulletin board still has items regarding that earthquake to this day.  Its commanding officer was by the name of Brooke Schumm (1905-1985) who commanded it from October 1952 to October 1953. 

Author Response

Reviewer Comment 1:

The calculation of the response rate needs clarification.  Using 520 as those who responded, an 80% response rate would mean that the original sample was 650 (?).  However, it is not clear how the sample analyzed appeared to be 502.  What happened to the other 18?  Did the final sample include both an adolescent and a parent?  Were there situations where one but not both responded?  It also seems unusual that the response rate for both parents and adolescents was 80% when the study presumably began with the same number of potential respondents/families and yet the final sample was 502 adolescents and 474 parents.  For example, on parental education, if that was obtained from the parents, the numbers add up to 500, more than 474.  Did that mean that parental education was reported by the adolescents or was that data available from other sources, like the schools?  These issues are probably not major but they should be cleared up.

Response to Reviewer:

We thank the reviewer for identifying the need for greater clarity regarding sample composition and response rates. We have revised Section 2.1 (“Participants”) to specify that 520 adolescent-parent dyads were initially recruited, reflecting an 80% response rate from an original pool of approximately 650 eligible families. Of these, 18 adolescent responses were excluded due to incomplete or invalid data, resulting in a final adolescent sample of 502. Similarly, 474 parent questionnaires were returned and complete; in 28 cases, adolescent data were retained without a corresponding parent form. Regarding parental education, this variable was primarily reported by parents; however, in cases where parent responses were missing, adolescents were asked to provide this information. These entries were used in descriptive statistics but not in analyses involving parent-reported clinical outcomes. These clarifications have been added to the manuscript to improve transparency.

Reviewer Comment 2:
The study would seem to have been approved in October 2014 but now it's April 2025. Any particular reason for the long delay in preparing these results? Furthermore, the earthquakes occurred in February 2014 and the study was on page 10 said to have been conducted six months after the earthquakes but the approval wasn't until October, which is eight to nine months after the earthquakes. Was approval obtained a couple of months after the study was begun? Not necessarily a big issue but something that needs clarification.

Response to Reviewer:
We thank the reviewer for highlighting this point. The date listed for ethical approval—10-02-2014—follows the European date format and refers to February 10, 2014, which was prior to the second earthquake (February 3, 2014). Thus, approval was obtained before data collection, and certainly not after. We have revised the Institutional Review Board Statement to specify the approval date in full (e.g., "10 February 2014") to avoid ambiguity.

As for the timeline of publication, while the study was conducted in October 2014, a combination of external factors—including administrative delays, a shift in research priorities during the COVID-19 pandemic, and subsequent internal quality reviews—contributed to the extended time before submission. These issues have now been addressed, and the current version reflects extensive revision and updates to ensure clarity and rigor. We have added a brief explanatory note in the limitations section to acknowledge this timeline.

Reviewer Comment 3:
As an historical side note, the USA ship, the USS Salem, was a heavy cruiser that helped the effected communities from the 1953 earthquake with food, water, and electricity.  That ship survives, unlike others that have been scrapped, as a museum ship in the Boston, USA area and its bulletin board still has items regarding that earthquake to this day.  Its commanding officer was by the name of Brooke Schumm (1905-1985) who commanded it from October 1952 to October 1953.

Response to Reviewer:

We sincerely appreciate the reviewer’s thoughtful and informative historical note regarding the humanitarian role of the USS Salem following the 1953 Ionian Sea earthquake. This is indeed a significant piece of history that underscores the longstanding international interest in disaster response and the legacy of support offered to Kefalonia. 

Reviewer 4 Report

Comments and Suggestions for Authors

Although this paper addresses a novel concept and timing of data given the twin earthquakes that were documented, the paper reports on data from 2014. Since then, there has been considerable advancement in technology and human behavior. The paper is well written, but I believe the data is too stale to make any meaningful contributions to the literature.

If the editors disagree, I would suggest adding the research instrument as Appendices to the study and connecting the results of the study to much more modern disasters, such as the 2025 California USA or the 2025 earthquakes in Thailand and Myanmar.

Author Response

Reviewer Comment 1:
Although this paper addresses a novel concept and timing of data given the twin earthquakes that were documented, the paper reports on data from 2014. Since then, there has been considerable advancement in technology and human behavior. The paper is well written, but I believe the data is too stale to make any meaningful contributions to the literature.

Response to Reviewer:

We respectfully acknowledge the reviewer’s concern regarding the age of the data. Although the study was conducted in 2014, we believe the findings remain highly relevant due to the enduring nature of the psychological mechanisms examined—particularly the interplay between disaster exposure, parental psychopathology, and youth outcomes. As noted in the revised manuscript (Section 4, Discussion), these mechanisms continue to manifest in contemporary disaster contexts. For example, while psychological data are still emerging, the 2025 Southern California wildfires and the 2025 Myanmar earthquake highlight the ongoing global need for evidence-based frameworks to guide post-disaster mental health interventions. Our study’s use of validated instruments, its ecological conceptual framework, and its inclusion of both parent and youth data offer enduring insights that can inform responses in current and future settings. Additionally, we have included a statement in the Discussion section to clarify the relevance of the findings.

Reviewer Comment 2:
“If the editors disagree, I would suggest adding the research instrument as Appendices to the study…”

Response:
We appreciate this suggestion. However, due to copyright restrictions and licensing agreements with the publishers of the psychometric instruments used (e.g., UCLA PTSD Reaction Index, IES-R, BDI-II, STAI), we are unable to reproduce the full instruments in the appendices. To maintain transparency and facilitate access, we have ensured that all instruments are fully cited with references and brief descriptions of their structure and scoring.

Round 2

Reviewer 1 Report

Comments and Suggestions for Authors

Dear authors, thank you very much for the changes; I just have one more observation. The reliability of The Screen for Children's Anxiety and Related Disorders–Short Form (SCARED-SF) is low, so please add this to the limitations.

Author Response

Reviewer 1 Comment:

Dear authors, thank you very much for the changes; I just have one more observation. The reliability of The Screen for Children's Anxiety and Related Disorders–Short Form (SCARED-SF) is low, so please add this to the limitations.

Response:

We thank the reviewer for this valuable observation. We have now added a statement to the limitations section (Section 4.1) noting that the internal consistency of the SCARED-SF in our sample was relatively low (Cronbach’s α = 0.64), which may have limited the reliability of anxiety symptom measurement.

Reviewer 2 Report

Comments and Suggestions for Authors

Dear Authors,

I am pleased with the corrections and revisions made by the authors in response to the feedback provided. The authors have effectively addressed the concerns raised, demonstrating a clear understanding of the issues and a commitment to improving the manuscript's quality.

Thank you.

Author Response

Reviewer 2 Comment:

I am pleased with the corrections and revisions made by the authors in response to the feedback provided. The authors have effectively addressed the concerns raised, demonstrating a clear understanding of the issues and a commitment to improving the manuscript's quality. Thank you.

Response:

We sincerely thank the reviewer for their kind and encouraging feedback. We appreciate your thoughtful comments throughout the review process, which have helped us substantially improve the clarity and quality of the manuscript.

Reviewer 4 Report

Comments and Suggestions for Authors

My original review stands--I believe the data is too old and the research instrument not being available is another issue. 

Author Response

Reviewer 4 Comment:

My original review stands--I believe the data is too old and the research instrument not being available is another issue.

Response:

We appreciate the reviewer’s continued engagement with our manuscript. While we understand concerns regarding the timing of data collection (2014), we respectfully note that the delay in publication was due to logistical and institutional constraints, including disruptions during the COVID-19 pandemic. These delays were transparently acknowledged in the Limitations section (Section 4.1). We also emphasize that the study offers novel, region-specific insights into family dynamics and adolescent trauma in post-disaster contexts that remain highly relevant for current and future crises, such as the 2025 Myanmar earthquake and Southern California wildfires. In summary, we believe the age of the data does not diminish its scientific contribution, especially given the rarity of large-scale, family-based earthquake studies in European settings. We remain confident that the findings provide meaningful implications for disaster mental health policy and practice.